# In Vitro Evaluation of Brown Seaweed *Laminaria* spp. as a Source of Antibacterial and Prebiotic Extracts That Could Modulate the Gastrointestinal Microbiota of Weaned Pigs

**DOI:** 10.3390/ani13050823

**Published:** 2023-02-24

**Authors:** Brigkita Venardou, John V. O’Doherty, Marco Garcia-Vaquero, Claire Kiely, Gaurav Rajauria, Mary J. McDonnell, Marion T. Ryan, Torres Sweeney

**Affiliations:** 1School of Veterinary Medicine, University College Dublin, Belfield, D04 V1W8 Dublin, Ireland; 2School of Agriculture and Food Science, University College Dublin, Belfield, D04 V1W8 Dublin, Ireland

**Keywords:** brown macroalgae, bifidogenic, antibacterial, seaweed polysaccharides, weaned pig, gastrointestinal microbiota, batch fermentation

## Abstract

**Simple Summary:**

The transition from milk to solid feed in commercial pig-production systems negatively affects gut health, particularly the composition of the residing microbial community. This can subsequently impair pig growth and long-term health. Natural dietary supplements including seaweed extracts have the capacity to reduce pathogen load (antibacterial activity) and/or increase beneficial microbes (prebiotic activity). This study evaluated the antibacterial and prebiotic potential of two seaweed species, *Laminaria hyperborea* and *Laminaria digitata,* and their extracts using laboratory-based simulations of the gut microbial community of weaned pigs. Our investigation identified seaweed extracts that could decrease the numbers of pig- and food-related pathogens or increase the number of beneficial microbes, albeit to a different extent. These findings indicate that seaweeds are a promising source of antibacterial and prebiotic dietary supplements for use in pigs during the weaning period.

**Abstract:**

*Laminaria* spp. and their extracts have preventative potential as dietary supplements during weaning in pigs. The first objective of this study was to evaluate increasing concentrations of four whole seaweed biomass samples from two different *Laminaria* species harvested in two different months in a weaned pig faecal batch fermentation assay. Particularly, February and November whole seaweed biomass samples of *L. hyperborea* (LHWB-F and LHWB-N) and *L. digitata* (LDWB-F and LDWB-N) were used. In the next part of the study, the increasing concentrations of four extracts produced from *L. hyperborea* (LHE1–4) and *L. digitata* (LDE1–4) were evaluated in individual pure-culture growth assays using a panel of beneficial and pathogenic bacterial strains (second objective). The LHE1–4 and LDE1–4 were obtained using different combinations of temperature, incubation time and volume of solvent within a hydrothermal-assisted extraction methodology (E1–4). In the batch fermentation assay, the *L. hyperborea* biomass samples, LHWB-F and LHWB-N, lowered *Bifidobacterium* spp. counts compared to the *L. digitata* biomass samples, LDWB-F and LDWB-N (*p* < 0.05). LHWB-F and LDWB-N reduced *Enterobacteriaceae* counts (*p* < 0.05). LHWB-F and LDWB-F were selected as the most and least promising sources of antibacterial extracts from which to produce LHE1–4 and LDE1–4. In the pure-culture growth assays, E1- and E4-produced extracts were predominantly associated with antibacterial and bifidogenic activities, respectively. LHE1 reduced both *Salmonella* Typhimurium and Enterotoxigenic *Escherichia coli* with LDE1 having a similar effect on both of these pathogenic strains, albeit to a lesser extent (*p* < 0.05). Both LHE1 and LDE1 reduced *B. thermophilum* counts (*p* < 0.05). LDE4 exhibited strong bifidogenic activity (*p* < 0.05), whereas LHE4 increased *Bifidobacterium thermophilum* and *Lactiplantibacillus plantarum* counts (*p* < 0.05). In conclusion, antibacterial and bifidogenic extracts of *Laminaria* spp. were identified in vitro with the potential to alleviate gastrointestinal dysbiosis in newly weaned pigs.

## 1. Introduction

A healthy gut microbiota which is compositionally and functionally diverse and stable is essential to support host health and growth [1,2,3,4]. Contrarily, dysbiosis represents a state of imbalance in the composition and function of this microbial community, characterised by decreases in beneficial microorganisms and/or overgrowth of pathogens and/or a loss of overall diversity, with a subsequent negative impact on gastrointestinal health [5]. Commercial weaning in pigs is a typical example of dysbiosis, whereby the transition from milk to solid feed, coupled with emotional, social and environmental stressors leads to gastrointestinal dysfunction characterised by dysbiosis that predisposes them to intestinal infection and disease [6,7]. 

In a recent review, the potential of marine macroalgae or seaweeds were considered as natural dietary supplements with which to promote gastrointestinal health and subsequently growth in weaned pigs [8]. Brown seaweeds are rich in nondigestible polysaccharides, minerals, polyphenols and vitamins [9,10]. Wide-ranging biological activities [11] have been attributed to seaweed components, particularly fucoidan and laminarin, including prebiotic [12,13] and antibacterial [14,15] potential. However, various factors influence the concentration, structure and biological activity of seaweed-derived polysaccharides, such as seaweed species, harvest season, environmental conditions, and extraction methodologies [14,16]. Recently, a multivariate statistic technique, response surface methodology, has been utilised to improve the extraction efficiency by optimising the extraction conditions for a selected seaweed polysaccharide and/or bioactivity [17]. In that study, a novel hydrothermal-assisted extraction (HAE) methodology with combinations of temperature, time and solvent to seaweed ratio optimised for the best concentration of laminarin and/or fucoidan and/or antioxidant activity was developed using the response surface methodology. Seaweed extracts of *Ascophyllum nodosum* produced using this HAE methodology exhibited enhanced antibacterial and prebiotic activity compared to the conventional extraction methods [18].

The brown seaweed *Laminaria* spp. is a rich source of biologically active nondigestible polysaccharides. Previous in vitro investigation has associated this seaweed species with various biological activities including anti-inflammatory, immunomodulatory, antioxidant, antitumor and antihypertensive [9,11] effects, several of which have also been observed in in vivo studies with pigs [19,20,21,22]. Concerning its effect on the gastrointestinal microbiota, dietary supplementation of pigs with crude *Laminaria* spp. extracts consistently led to a reduction in the numbers of the *Enterobacteriaceae* family [22,23,24,25,26,27] which include several animal and human pathogens such as *Salmonella enterica* subsp. *enterica* serotype Typhimurium and pathogenic *Escherichia coli* [28,29]. Furthermore, an increase in *Enterobacteriaceae* family is considered to be an indication of dysbiosis and a risk factor for post-weaning diarrhoea in pigs [6,30]. Dietary supplementation of pigs with crude *Laminaria* spp. extracts led to a more variable response in the intestinal lactobacilli and *Bifidobacterium* spp. populations, as both increases [22] and decreases [23,24] in their counts have been reported. Lactobacilli are dominant members of the gastrointestinal microbiota in pigs and have an important role in growth and health due to their contributions to nutrient bioavailability, inhibition of pathogen colonisation and immunomodulation [31,32]. *Bifidobacterium* spp. are considered a beneficial bacterial population due to their probiotic status [33], but are present in low abundance in the gastrointestinal tract of pigs [34]. 

The current study focused on the antibacterial and prebiotic potential of two polysaccharide-rich members of the *Laminaria* spp., *L. digitata* and *L. hyperborea*, with an average total carbohydrate content of 70.7% and 65.5% of dry weight, respectively, as described in previous reports [35]. Batch fermentation and pure-culture growth assays were useful screening tools when assessing the direct effects of whole biomass seaweed samples and their extracts on key bacterial populations and species in the porcine gastrointestinal tract [18]. Thus, the first objective of this study was to assess the influence of seaweed species and harvest season on the effect of whole biomass samples of *L. digitata* and *L. hyperborea* with respect to selected faecal bacterial populations in a batch fermentation assay inoculated with pig faeces. The second objective was to investigate whether the different extraction conditions of the HAE methodology led to *L. digitata* and *L. hyperborea* extracts with improved antibacterial and prebiotic activities using a panel of pure-culture growth assays.

## 2. Materials and Methods

### 2.1. Laminaria spp.: Whole Biomass Samples and Extracts

The whole biomass samples (WB) of *L. digitata* (LD) and *L. hyperborea* (LH) were harvested in February (LDWB-F and LHWB-F) and November (LDWB-N and LHWB-N) by Quality Sea Veg Ltd., Co. (Donegal, Ireland). For each seaweed species, whole biomass samples were collected at a single time point and from the same collection site. The preparation (oven-dying, milling) and compositional analysis (dry matter, ash, protein, crude lipids, polysaccharide content, total phenols) of the dried whole seaweed biomass samples was performed as previously described [36]. LDWB-F, LDWB-N, LHWB-F and LHWB-N were stored at room temperature until their evaluation in the batch fermentation assay. 

A HAE methodology with optimised extraction conditions (temperature, incubation time and volume of solvent) was used to produce the extracts of LDWB-F and LHWB-F, as described previously by Garcia-Vaquero et al. [17] and presented in Table 1. The parameters for each extraction condition were optimised towards the concentration of fucoidan for E1, laminarin for E2, antioxidant activity for E3 and all the above for E4. The produced extracts of *L. digitata* (LDE1–4) and *L. hyperborea* (LHE1–4) were freeze-dried and their laminarin and fucoidan content was determined as previously described [18]. All extracts were analysed on two independent occasions (two biological replicates) with three readings each time. LDE1–4 and LHE1–4 were stored at −20 °C until their evaluation in the pure-culture growth assays.

### 2.2. Batch Fermentation Assay

The preparation of the faecal inoculum and the batch fermentation assay were carried out as described previously [37]. Briefly, faeces from 29 healthy newly weaned crossbred pigs (Large White × Landrace) fed a cereal- and milk-based diet were pooled, aliquoted and stored at −20 °C. One day prior to the batch fermentation assay, the pooled faeces were diluted (1:5 w/v) in phosphate-buffered saline (Sigma-Aldrich, St. Louis, MO, USA) after oxygen removal using oxyrase (Sigma-Aldrich, St. Louis, MO, USA) to prepare the faecal inoculum (FI) that was stored at 4 °C anaerobically. The FI was added to the fermentation medium at a 1:10 v/v ratio (21 mL final volume). The inclusion levels of LDWB-F, LHWB-F, LDWB-N and LHWB-N in the FI/fermentation medium were 0 (control tubes), 1, 2.5 and 5 mg/mL. The batch fermentation was carried out under anaerobic conditions using oxyrase and CO_2_ flushing at 39 °C for 24 h with gentle stirring (100 rpm). Sampling (5 mL fermentation broth) was performed at 0 and 24 h in duplicate. After centrifuging at 12,000× *g* for 5 min, the resultant pellets were stored in −20 °C until further analysis. All experiments were repeated on three independent occasions (biological replicates *n* = 3). 

### 2.3. Quantification of Bacterial Groups Using Quantitative Real Time Polymerase Chain Reaction (QPCR)

DNA extraction: Bacterial DNA was extracted using QIAamp Fast DNA stool mini kit (Qiagen, West Sussex, UK) according to the manufacturer’s instructions, and its quantity and quality was evaluated spectrophotometrically (Nanodrop, Thermo Fisher Scientific, Waltham, MA, USA).

Bacterial primers: The primers targeting the 16S rRNA gene of selected bacterial groups (total bacteria, lactobacilli and *Bifidobacterium* spp.) or the *rplP* gene (*Enterobacteriaceae*) are provided in Table 2. Primer design software, Primer3 (https://primer3.org/ (accessed on 26 June 2018)) and Primer Express™ (Applied Biosystems, Foster City, CA, USA) were used for larger amplicons (>150 bp) and smaller amplicons (<125 bp), respectively. Primer specificity was verified using Primer Basic Local Alignment Search Tool (Primer-BLAST), https://www.ncbi.nlm.nih.gov/tools/primer-blast/index.cgi, accessed on 26 June 2018.

Bacterial enumeration by QPCR: The quantification of the above-mentioned bacterial groups was carried out using QPCR with plasmid-based standard curves as described previously [37]. Briefly, Competent *E. coli* was transformed with a pCR4-TOPO™ TA vector containing each fragment of the targeted 16S rRNA genes for total bacteria, lactobacilli and Bifidobacterium spp. or the rplP gene for *Enterobacteriaceae* and the resistance to ampicillin gene using a TOPO™ TA Cloning™ Kit for Sequencing (Invitrogen, Thermo Fisher Scientific, Carlsbad, CA, USA) and stored in cryoprotective beads (TS/71-MX, Protect Multi-purpose, Technical Service Consultants Ltd., Lancashire, UK). Transformed *E. coli* was recultured in 200 mL LB Broth Base (Invitrogen, Thermo Fisher Scientific, Carlsbad, CA, USA) containing ampicillin (100 µg/mL) at 37 °C for 18 h at 150 rpm. Plasmids were extracted on a large scale using the GenElute™ HP Plasmid Maxiprep kit, (Sigma-Aldrich, St. Louis, MO, USA), linearised using APA1 restriction enzyme (Promega, Madison, WI, USA) and purified using GenElute™ PCR Clean-Up kit (Sigma-Aldrich, St. Louis, MO, USA) according to the manufacturers’ instructions. Following quantification, the plasmid copy number/μL was determined using the URI Genomics & Sequencing Centre online tool (http://cels.uri.edu/gsc/cndna.htmL, accessed on 14 May 2019). For the QPCR, the final reaction volume (20 μL) included 3 μL template DNA, 1 μL of forward primer (10 μM), 1 μL of reverse primer (10 μM), 5 μL nuclease-free water and 10 μL of Fast SYBR^®^ Green Master Mix (Applied Biosystems, Foster City, CA, USA) for the lactobacilli or GoTaq^®^ qPCR Master Mix (Promega, Madison, WI, USA) for the remaining bacterial groups. All QPCR reactions were performed in duplicate on the ABI 7500 Fast PCR System (Applied Biosystems, Foster City, CA, USA) using the following cycling conditions: a denaturation step (95 °C/10 min), 40 cycles of 95 °C for 15 s and 60 °C for 1 min. Dissociation curve analysis and visualisation on a 2% agarose gel stained with ethidium bromide were used to confirm the production of single and specific PCR products. All PCR reactions used in this study exhibited 90–110% efficiency established by plotting the threshold cycles (Ct) derived from the 5-fold serial dilutions of each plasmid against their arbitrary quantities. Bacterial counts were determined using the standard curve derived from the mean Ct value and the log-transformed gene copy number of the respective plasmid and expressed as log-transformed gene copy number per gram of digesta (logGCN/g digesta).
animals-13-00823-t002_Table 2Table 2List of forward and reverse primers used for the bacterial quantification by QPCR.Target Bacterial GroupForward Primer (5′–3′)Reverse Primer (5′–3′)Amplicon Length (bp) Tm (°C)ReferencesTotal bacteriaF: GTGCCAGCMGCCGCGGTAAR: GACTACCAGGGTATCTAAT29164.252.4[38]LactobacilliF: AGCAGTAGGGAATCTTCCAR: CACCGCTACACATGGAG34154.555.2[39]*Bifidobacterium* spp.F: GCGTGCTTAACACATGCAAGTCR: CACCCGTTTCCAGGAGCTATT12560.359.8[40]*Enterobacteriaceae*F: ATGTTACAACCAAAGCGTACAR: TTACCYTGACGCTTAACTGC18554.056.3[41]bp, base pairs; Tm, melting temperature.

### 2.4. Bacterial Strains and Pure-Culture Growth Assays

Pure-culture growth assays using a panel of commensal strains *Lactiplantibacillus plantarum* subsp. *plantarum* (formerly *Lactobacillus plantarum*, DSMZ 20174), *Limosilactobacillus reuteri* (formerly *Lactobacillus reuteri*, DSMZ 20016) and *Bifidobacterium thermophilum* (DSMZ 20210) and pathogens *S. typhimurium* PT12 and enterotoxigenic *E. coli* (ETEC) O149A+ selected for their beneficial roles and negative impacts on pig and human health, respectively, were carried out as described in our previous work [37]. Briefly, 24 h cultures of all bacterial strains were prepared using standard procedures and diluted in 10% medium: 10% de Man, Rogosa and Sharpe broth (MRS, Oxoid Ltd., Hampshire, UK) for *L. plantarum*, *L. reuteri* and *B. thermophilum* cultures; or 10% Tryptone Soya Broth (TSB, Oxoid Ltd., Hampshire, UK) for *S. typhimurium* and ETEC cultures, to obtain an inoculum of 10^6^–10^7^ CFU (colony-forming unit)/mL (verified for each assay). Two-fold dilutions (2–0.125 mg/mL) of LDE1–4 and LHE1–4 were performed in 10% MRS and 10% TSB prior to each assay from a working concentration of 4 mg/mL. 100 μL of each extract and each dilution and 100 μL of inoculum were added to duplicate wells of 96-well microtiter plates (CELLSTAR, Greiner Bio-One, Kremsmünster, Austria). Control wells were also included (bacterial inoculum only). Assay sterility was assessed using blank wells (no bacterial inoculum) for each dilution of each extract. After gentle agitation to ensure thorough mixing, plates were incubated aerobically at 37 °C for 18 h, apart from *B. thermophilum*, which was incubated anaerobically. Afterwards, bacterial enumeration was carried out by 10-fold serial dilution (10^−1^–10^−8^), spread plating onto MRS agar (Oxoid Ltd., Hampshire, UK) for *L. plantarum*, *L. reuteri* and *B. thermophilum*, and Tryptone Soya Agar (Oxoid Ltd., Hampshire, UK) for ETEC and *S. typhimurium*, and incubation aerobically at 37 °C for 24 h or anaerobically at 37 °C for 48 h for *B. thermophilum*. The dilution resulting in 5–50 colonies was selected for the calculation of CFU/mL using the formula, CFU/mL = average colony number × 50 × dilution factor. The bacterial counts were logarithmically transformed (logCFU/mL) for the subsequent statistical analysis. All experiments were carried out with technical replicates on three independent occasions (3 biological replicates *n*).

### 2.5. Statistical Analysis

All data were statistically analysed using Statistical Analysis Software (SAS) 9.4 (SAS Institute, Cary, NC, USA). Normality tests were initially carried out using PROC UNIVARIATE procedure for each data set. 

*Batch fermentation assay:* For each bacterial group and tested compound, the bacterial counts (*n* = 12, 3 flasks/each compound concentration) were analysed using PROC GLM procedure (Tukey’s test). The statistical model included the fixed effects of seaweed species (*L. digitata*, *L. hyperborea*), the season of collection (February, November), the concentration of whole biomass (0, 1, 2.5 and 5 mg/mL) and assay replicates (3 biological replicates) and their associated two- and three-way interactions with the bacterial counts at 0 h as a covariate. 

*Pure-culture growth assay:* To control for the natural variability in bacterial growth in the pure-culture assays, bacterial counts were expressed as the difference between the counts of each bacterial strain for each extract concentration and their respective control (0 mg/mL). The resulting positive or negative values representing the difference in bacterial counts were analysed using PROC GLM procedure (Tukey’s test). The statistical model assessed the effects of seaweed species (*L. digitata*, *L. hyperborea*), extraction conditions (E1–4) and concentration of extracts (0.125, 0.25, 0.5, 1 and 2 mg/mL) and their associated two- and three-way interactions. The biological replicate was the experimental unit.

Probability values of < 0.05 denote statistical significance. Results are presented as least-square mean values ± standard error of the means (SEM).

## 3. Results

### 3.1. Proximate Composition of L. digitata and L. hyperborea, and Laminarin and Fucoidan Content of Their Extracts

The proximate composition of the whole seaweed biomass samples LDWB-F, LHWB-F, LDWB-N and LHWB-N is presented in Table 3, as reported previously [36]. 

The laminarin and fucoidan contents in the *L. digitata* (LDE1–4) and *L. hyperborea* (LHE1–4) extracts are presented in Table 4.

### 3.2. Effects of the Whole Biomass Samples of L. digitata and L. hyperborea on Selected Bacterial Populations

The effects of the whole biomass samples of *L. digitata* and *L. hyperborea* collected in February (LDWB-F and LHWB-F) and November (LDWB-N and LHWB-N) were evaluated on selected faecal bacterial populations in a batch fermentation assay. The effects of species, season and concentration and their interactions are presented in Table 5 and Table 6 and are described below. The species × concentration interaction and the season × concentration interaction were only significant for *Bifidobacterium* spp. (*p* < 0.05) and, as a result, were excluded from the statistical analysis of the other bacterial groups.

*Enterobacteriaceae*: There was a species × season interaction whereby LHWB-F led to lower *Enterobacteriaceae* counts compared to LHWB-N, while the opposite was true for *L. digitata* (*p* < 0.05, Table 5). There was also a concentration effect on *Enterobacteriaceae* counts, whereby the 5 mg/mL reduced *Enterobacteriaceae* counts compared to the control (7.74 logGCN/g digesta (5 mg/mL) vs. 7.99 logGCN/g digesta (0 mg/mL) ± 0.056, *p* < 0.05).

*Bifidobacterium* spp.: There was a species × season interaction, whereby LHWB-N led to lower *Bifidobacterium* spp. counts compared to LHWB-F, while harvest season had no effect on *Bifidobacterium* spp. counts with regard to *L. digitata* (*p* > 0.05, Table 5). There was a species × concentration interaction, whereby the concentrations of 2.5 and 5 mg/mL of *L. hyperborea* led to lower *Bifidobacterium* spp. counts compared to the control and 1 mg/mL (2.73 logGCN/g digesta (2.5 mg/mL) and below the limits of detection (5 mg/mL) vs. 6.72 (0 mg/mL) and 6.60 (1 mg/mL) logGCN/g digesta ± 0.056, *p* < 0.05)), while this effect was not as potent with the corresponding *L. digitata* concentrations (5.95 logGCN/g digesta (5 mg/mL) vs. 6.46 (0 mg/mL), 6.55 (1 mg/mL) and 6.37 (2.5 mg/mL) logGCN/g digesta ± 0.056, *p* < 0.05)). There was a season × concentration interaction, whereby the 5 mg/mL of both February and November suppressed *Bifidobacterium* spp. counts, while at 2.5 mg/mL, there was a greater reduction in the counts in November relative to February (*p* < 0.05, Table 6). 

Total bacteria: There was a species effect and a concentration effect on total bacterial counts. *L. hyperborea* increased total bacteria compared to *L. digitata* (9.83 logGCN/g digesta (*L. hyperborea*) vs. 9.72 logGCN/g digesta (*L. digitata*) ± 0.034, *p* < 0.05). The 1 and 2.5 mg/mL gave higher counts compared to the control (9.87 (1 mg/mL) and 9.87 (2.5 mg/mL) logGCN/g digesta vs. 9.64 logGCN/g digesta (0 mg/ mL) ± 0.048, *p* < 0.05).

Lactobacilli: There was a species effect and a concentration effect on lactobacilli counts. *L. hyperborea* increased lactobacilli counts compared to *L. digitata* (8.79 logGCN/g digesta (*L. hyperborea*) vs. 8.45 logGCN/g digesta (*L. digitata*) ± 0.030, *p* < 0.05). The concentrations of 1 and 2.5 mg/mL were associated with higher counts compared to the control (8.66 logGCN/g digesta (1 mg/mL) and 8.69 logGCN/g digesta (2.5 mg/mL) vs. 8.56 logGCN/g digesta (0 mg/mL) ± 0.035, *p* < 0.05).

In summary, whole seaweed biomass samples from *L. hyperborea* and *L. digitata* collected in February had the least negative impact on *Bifidobacterium* spp. counts. Furthermore, LHWB-F reduced *Enterobacteriaceae* counts to the greatest degree, while LDWB-F had no effect. Both LHWB-F and LDWB-F were selected to generate the extracts evaluated in the next part of the screening process to determine whether the extraction methodology could improve the bioactivity of two whole seaweed biomass samples with varying effects. 

### 3.3. Identifying L. digitata and L. hyperborea Extracts with the Highest Antibacterial and Prebiotic Potential in Pure Bacterial Cultures

The HAE methodology with four different extraction conditions (E1–4) was employed for producing the extracts from the *L. digitata* (LD) and *L. hyperborea* (LH) samples, collected in February, to investigate whether the extraction method could improve their biological properties. LDE1–4 and LHE1–4 were evaluated for their antibacterial and prebiotic activities in pure-culture growth assays with selected beneficial (*L. plantarum*, *L. reuteri*, *B. thermophilum*) and pathogenic (ETEC, *S. typhimurium*) bacterial strains. Bacterial counts were expressed as the difference between the counts of each bacterial strain for each extract concentration and their respective control (0 mg/mL). The effects of species, extraction condition and concentration and their interactions are presented in Table 7 and Table 8 and are described below. The species × concentration interaction was significant only for *S. typhimurium* (*p* < 0.05) and was excluded from the statistical analysis of the other bacterial species.

#### 3.3.1. The Effect of the Different Extraction Conditions on the Antibacterial and Prebiotic Effects of *L. hyperborea* and *L. digitata* Extracts

ETEC and *S. typhimurium*: There was a species × extraction condition interaction, whereby LHE1 had more potent antibacterial activity than LHE2, LHE3 and LHE4, whereas the effect of the E1 extraction condition was not as potent with *L. digitata*, despite being significant (*p* < 0.05, Table 7). 

*B. thermophilum*: There was a species × extraction condition interaction, whereby LDE4 was more bifidogenic than LDE1, LDE2 and LDE3, whereas the effect of E4 extraction condition was not as evident with *L. hyperborea*, despite being significant compared with E1 and E2 (*p* < 0.05, Table 7). 

*L. plantarum*: There was a species × extraction condition interaction, whereby LHE4 was more stimulating on *L. plantarum* growth than LHE1, LHE2 and LHE3 (*p* < 0.05) and there was no effect of the extraction condition on *L. digitata* (*p* > 0.05, Table 7). 

*L. reuteri*: There was a species effect and an extraction condition effect on *L. reuteri* counts. LH extracts led to higher *L. reuteri* counts compared to LD extracts (0.24 logCFU/mL (LH extracts) vs. 0.16 logCFU/mL (LD extracts) ± 0.029, *p* < 0.05). The extraction conditions E1 and E2 increased *L. reuteri* counts compared to E3 (0.30 logCFU/mL (E1) and 0.27 logCFU/mL (E2) vs. 0.07 logCFU/mL (E3) ± 0.041, *p* < 0.05).

#### 3.3.2. The Effect of Concentration on the Antibacterial and Prebiotic Activity of the Different Extraction Conditions

ETEC and *S. typhimurium*: There was a concentration × extraction condition interaction, whereby 2 mg/mL was more potent than 1, 0.5, 0.25 and 0.125 mg/mL for the E1 extraction condition (*p* < 0.05), whereas the effect of concentration was not as evident with E2, E3 and E4 extraction conditions (*p* > 0.05, Table 8). There was also a species × concentration interaction for *S. typhimurium* (*p* < 0.05), whereby the 2 mg/mL LH extracts led to lower counts compared to the 2 mg/mL LD extracts (−1.56 logCFU/mL (LH extracts) vs. −0.66 logCFU/mL (LD extracts) ± 0.058, *p* < 0.05), however, species had no effect at any of the other concentrations.

*B. thermophilum*: There was a concentration × extraction condition interaction, whereby all concentrations of E4 were more bifidogenic than the equivalent concentrations in E1, E2 and E3, where some of the concentrations had no effect, while some were antibacterial (*p* < 0.05, Table 8). 

*L. plantarum*: There was a concentration effect on *L. plantarum* counts. The concentration of 1 and 2 mg/mL of all extracts increased *L. plantarum* counts compared to the 0.125 mg/mL (0.15 (1 mg/mL) and 0.15 (2 mg/mL) logCFU/mL vs. 0.02 logCFU/mL (0.125 mg/mL) ± 0.032, *p* < 0.05). 

*L. reuteri*: There was a concentration effect on *L. reuteri* counts. The concentration of 2 mg/mL of all extracts increased *L. reuteri* counts compared to 0.125 mg/mL (0.34 logCFU/mL (2 mg/mL) vs. 0.10 logCFU/mL (0.125 mg/mL) ± 0.045, *p* < 0.05). 

## 4. Discussion

The influence of seaweed species and harvest season on the effects of the whole biomass samples of *L. hyperborea* and *L. digitata* on selected bacterial markers of the porcine faecal microbiota were evaluated in a batch fermentation assay. In this study, seaweed species was the predominant factor affecting the growth of *Bifidobacterium* spp., *Enterobacteriaceae*, lactobacilli and total bacteria. *Bifidobacterium* spp. counts were also influenced by the harvest season. The February-harvested *L. hyperborea* biomass sample, LHWB-F, led to the lowest *Enterobacteriaceae* counts among all tested samples whilst also having a reduced negative impact on *Bifidobacterium* spp. compared to the November-harvested counterpart, LHWB-N. Contrarily, the February-harvested *L. digitata* biomass sample, LDWB-F, was the least promising in terms of its antibacterial properties, having no major effects on the tested bacterial groups. These two whole biomass seaweed samples were used to produce LHE1–4 and LDE1–4 using four extraction conditions (E1–4) of the HAE methodology. The extracts were assessed in a panel of pure-culture growth assays with selected beneficial and pathogenic bacterial strains, to evaluate whether the optimised extraction conditions could enhance their antibacterial and prebiotic activities. Regardless of the seaweed species, the extraction condition E1 was predominantly associated with improved antibacterial activity against *S. typhimurium*, ETEC and to a lesser extent *B. thermophilum*, while the E4 extraction condition was predominantly associated with bifidogenic activity. 

Total bacteria, lactobacilli, *Bifidobacterium* spp. and *Enterobacteriaceae* were monitored in the batch fermentation assay as part of the evaluation of the whole biomass of *L. digitata* and *L. hyperborea* collected in February (LDWB-F and LHWB-F) and November (LDWB-N and LHWB-N). The whole biomass of *L. hyperborea*, LHWB-N and LHWB-F, reduced the *Bifidobacterium* spp. counts in a concentration-dependent manner with LHWB-F having a lesser impact. The whole biomass of *L. digitata*, LDWB-F and LDWB-N, also showed evidence of minor reductions in this bacterial population. In addition, LHWB-F and LDWB-N were associated with reduced *Enterobacteriaceae* counts. Reductions in *Bifidobacterium* spp. and *Enterobacteriaceae* counts have been previously observed in the faeces and colonic and caecal digesta in pigs supplemented with crude extracts of *L. hyperborea, L. digitata* or *Laminaria* spp. [23,24,25,26]. In this study, whole biomass samples of *L. hyperborea* were associated with minor increases in lactobacilli and total bacterial counts compared to whole biomass samples of *L. digitata*. Thus, bacterial growth was predominantly influenced by the seaweed species rather than harvest season, which only had a significant effect on the *Bifidobacterium* spp. population. 

It is interesting to hypothesise what the bioactive components within the whole seaweed biomass samples could be based on their proximate composition analysis and the results of the batch fermentation assay. The whole biomass samples of *L. hyperborea* had higher total polysaccharide content compared to *L. digitata* for both months. The main polysaccharides present in *L. digitata* and *L. hyperborea* are laminarin, mannitol, alginate and cellulose, of which laminarin and mannitol have been reported to exhibit significant seasonal variation in their concentration [35]. This, along with the increase in total glucans (laminarin and cellulose combined) observed in November for both seaweed species in the current study suggests that the variation in the total carbohydrate content was due to laminarin. Fucoidan was confirmed to be a relatively minor polysaccharide in the whole biomass samples of *L. digitata* and *L. hyperborea* as expected for the *Laminaria* spp. [42] and increased in November in both seaweed species. Previous research has reported that laminarin reduced the *Enterobacteriaceae* counts in the caecum and increased lactobacilli counts in the faeces and colon of weaned pigs [22,27,43], while fucoidan from whole *A. nodosum* biomass samples was considered to be the bioactive reducing *Bifidobacterium* spp. and *Enterobacteriaceae* counts in a batch fermentation assay inoculated with faeces from weaned pigs [18]. The reduction in *Bifidobacterium* spp. counts could additionally be attributed to inhibitory effects due to the wide-ranging components within the extracts, including phenols, alginate, cellulose and fucoidan, on the activity of bacterial carbohydrate-degrading enzymes [44,45,46]. As whole seaweed biomass samples are inherently complex, it is not possible to attribute the observed effects on the faecal microbiota to specific bioactive components within the whole biomass samples of *L. hyperborea* and *L. digitata* with certainty.

For the second part of the study, LHE1–4 and LDE1–4 were produced from LHWB-F and LDWB-F, respectively, using the HAE methodology with four extraction conditions (E1–4). Of these, LHWB-F was identified as the most promising antibacterial sample in the batch fermentation assay and was selected for further analysis. In parallel, LDWB-F was included to investigate whether the extraction protocol could improve its limited bioactivity, an effect that was demonstrated in a previous study [18]. LHE1–4 and LDE1–4 were evaluated for their antibacterial and prebiotic potential in a panel of pure-culture growth assays. The pathogens *S. typhimurium* and ETEC were selected as representatives of the *Enterobacteriaceae* family. While *S. typhimurium* infection in pigs is mostly asymptomatic, it is associated with intestinal inflammation and compositional changes in the gastrointestinal microbiota that can have a negative impact on animal health and performance [47,48,49]. Furthermore, pigs and their meat products can become a reservoir for *S. typhimurium*, which can impact on human health [50]. ETEC infection in newly weaned pigs contributes to the development of post-weaning diarrhoea, an economically significant disease characterised by diarrhoea, dehydration, stunted growth and significant mortality [51]. The effects of the *L. hyperborea* and *L. digitata* extracts on representative beneficial bacterial strains, *B. thermophilum, L. plantarum* and *L. reuteri*, were also evaluated. These bacterial species commonly colonise the porcine gastrointestinal tract and exert a range of beneficial roles such as inhibition of intestinal pathogens, immunomodulation, improved composition in the gastrointestinal microbiota and enhanced health and growth [52,53,54,55,56,57,58]. In the pure-culture growth assays, the E1 and E4 extraction conditions were predominantly associated with antibacterial and bifidogenic activities, respectively. LHE1 was the most potent extract in reducing *S. typhimurium* and ETEC counts. LDE1 also inhibited the growth of both pathogenic strains to a lesser extent. Additionally, LHE1 and LDE1 reduced *B. thermophilum* counts, whereas both extracts were also associated with a slight increase in *L. reuteri* counts. Interestingly, LDE4 followed by LHE4 increased *B. thermophilum* counts in a concentration-dependent manner, with LHE4 additionally stimulating the growth of *L. plantarum*. Based on the above, the use of the E1 and E4 extraction conditions of the HAE methodology produced antibacterial and bifidogenic extracts with the potential to promote a healthy composition in the gastrointestinal microbiota of pigs. 

The laminarin and fucoidan contents of LHE1–4 and LDE1–4 were determined to establish the concentrations of these polysaccharides achieved by each combination of extraction conditions of the HAE methodology [17]. LHE1–4 extracts had higher concentrations of laminarin and fucoidan compared to LDE1–4, an expected outcome based on the proximate composition of the respective whole seaweed biomass. Interestingly, both sets of extracts had higher fucoidan content (12.76–14.68% for LHE1–4 and 3.84–5.80% for LDE1–4) than laminarin content (4.94–7.59% for LHE1–4 and ≤0.70% for LDE1–4). The presence of laminarin is reported to be at lower concentrations during the winter months in these seaweed species, in agreement with our observation [35,59]. Apart from laminarin and fucoidan, alginate is a polysaccharide which is present in high and relative stable concentrations throughout the year in both *L. hyperborea* and *L. digitata* [35], and could also be a significant component of the LHE1–4 and LDE1–4. While the alginate content of the tested extracts was not determined in the current study, this assumption is supported by the findings of a recent study evaluating an *L. hyperborea* extract produced using the E2 extraction conditions of HAE methodology [60]. Furthermore, the different extraction conditions (Table 1) could affect not only the content but also the structure of these seaweed polysaccharides in the produced extracts, and hence, their bioactivity. For instance, the use of HCl and increasing temperatures in the extraction protocol was previously associated with changes in the chemical composition (monosaccharide content, sulphation level) and lower molecular weight due to partial hydrolysis of fucoidan and partial depolymerisation of alginate [61,62,63]. 

Although we did not determine the antibacterial and bifidogenic components of the *L. hyperborea* and *L. digitata* E1 and E4 extracts, we hypothesise that fucoidan was likely the main bioactive, with the variation in bioactivities attributed to structural alterations due to the different extraction conditions (Table 1). Regarding the antibacterial activity, this assumption is supported by the following three facts: (1) LHE1 had both higher fucoidan content and stronger antibacterial activity against *S. typhimurium* and ETEC compared to LDE1, suggesting a connection between this bioactivity and fucoidan; (2) The fucoidan-rich *A. nodosum* extracts produced using the same E1 extraction protocol also led to significant reductions in *S. typhimurium* and ETEC counts in our previous studies [18,64]; (3) Depolymerised fucoidans from *Laminaria* spp., *Sargassum* spp. and *Undaria* spp. were reported to have improved antibacterial activity against various pathogenic strains including *E. coli* and *S*. *typhimurium* compared to the parent polysaccharide [65,66,67]. The antibacterial activity of LHE1 and LDE1 against *B. thermophilum* indicate that bioactives other than fucoidan are involved. The bifidogenic effect of LHE4 and LDE4 may also be attributed to the depolymerised fucoidan fraction due to the similar effects on *Bifidobacterium* spp. growth of the fucoidan-rich *A. nodosum* extract produced using the same E4 extraction protocol and depolymerised fucoidans of *Laminaria* spp. and *Sargassum* spp. in previous in vitro studies [18,68,69]. Alginate oligosaccharides have also exhibited a bifidogenic effect in pure-culture growth assays [70,71]. Therefore, depolymerised alginate may have contributed to the increases in *B. thermophilum*, particularly in the case of LDE4. The slight increases in *L. plantarum* and *L. reuteri* counts with LHE4 counts and E1-produced extracts, respectively, indicate limited ability of these bacterial strains to utilise seaweed polysaccharides, most likely laminarin [72] and alginate oligosaccharides [71]. Taken together, all of the above results suggest a strong indication that fucoidan is the candidate bioactive responsible for the antibacterial and bifidogenic activities, although other seaweed constituents such as alginate may also contribute to the latter in the E4-produced extracts, particularly LDE4. In future studies, investigation into the chemical composition of LHE1, LDE1, LHE4 and LDE4 would provide better insight into the prebiotic and antibacterial bioactive components of these extracts, which was not possible at the laboratory-scale production of the extracts during the development of the HAE methodology.

## 5. Conclusions

The species of seaweed was the main determinant of the growth of *Bifidobacterium* spp., *Enterobacteriaceae* and lactobacilli when whole seaweed biomass samples were tested in a porcine batch fermentation assay. Whole biomass samples of *L. hyperborea* (LHWB-F) and *L. digitata* (LDWB-F) harvested in February were then selected as the most and least promising sources, respectively, for the generation of antibacterial extracts, based on their effects on the *Enterobacteriaceae* counts in the batch fermentation assay. E1- and E4-produced extracts from both seaweed species were associated with antibacterial and bifidogenic activities, respectively, indicating that the extraction conditions were a more important determinant of bioactivity than seaweed species. Of these extracts, LHE1 was the most potent extract against *S. typhimurium* and ETEC, whereas LDE4 stimulated the growth of *B. thermophilum* to the greatest extent. Further compositional characterisation of these extracts is required to facilitate the identification and purification of the bioactive components involved in the observed bioactivities. Nevertheless. these crude extracts, particularly LHE1, merit further exploration in terms of their ability to promote a more beneficial microbiota and, thus, overall health and growth in weaned pigs, as a means of minimising the costs associated with the purification of the responsible bioactives from these extracts.

## Figures and Tables

**Table 1 animals-13-00823-t001:** Extraction conditions employed to obtain the different *L. digitata* and *L. hyperborea* extracts.

*Laminaria* spp. Extract	Extraction Method *	Solvent *	Extraction Conditions *	Optimised for Targeted Bioactives
LDE1 LHE1	HAE	0.1 M HCl	120 °C 62.1 min 30 mL solvent/g seaweed	Fucoidan
LDE2 LHE2	HAE	0.1 M HCl	99.3 °C 30 min 21.3 mL solvent/g seaweed	Laminarin
LDE3 LHE3	HAE	0.1 M HCl	120 °C 76.06 min 10 mL solvent/g seaweed	Antioxidant activity
LDE4 LHE4	HAE	0.1 M HCl	120 °C 80.9 min 12.02 mL solvent/g seaweed	For laminarin, fucoidan and antioxidant activity

* Extraction methods and conditions used to produce *L. digitata* and *L. hyperborea* extracts as described by Garcia-Vaquero et al. [17]. LDE1–4, *L. digitata* extract 1–4; LHE1–4, *L. hyperborea* extract 1–4; HAE, hydrothermal-assisted extraction.

**Table 3 animals-13-00823-t003:** Proximate composition (dry matter, ash, protein, crude lipids, total glucans, fucose and phenols) of whole *L. digitata and L. hyperborea* biomass (data published in [36]).

Proximate Composition *	Whole *L. digitata* Biomass		Whole *L. hyperborea* Biomass	
	LDWB-F	LDWB-N	LHWB-F	LHWB-N
Dry matter (%)	91.39 ± 0.01	95.93 ± 0.01	90.83 ± 0.00	95.75 ± 0.02
Ash (% DW basis)	34.84 ± 0.08	21.82 ± 0.00	30.01 ± 0.03	18.91 ± 0.16
Protein (% DW basis)	11.12 ± 0.76	4.01 ± 0.04	9.98 ± 0.01	3.57 ± 0.00
Ether extract (% DW basis)	0.26 ± 0.05	1.12 ± 0.05	0.76 ± 0.07	0.69 ± 0.06
Total soluble sugars (% DW basis)	11.88 ± 0.13	20.39 ± 0.56	14.49 ± 0.11	26.69 ± 0.05
Total glucans (% DW basis)	1.51 ± 0.02	17.68 ± 0.09	6.40 ± 0.09	25.70 ± 0.10
Fucose (% DW basis)	0.77 ± 0.09	4.83 ± 0.15	2.66 ± 0.03	4.86 ± 0.05
Total phenolic content (% DW basis)	0.06 ± 0.00	0.05 ± 0.00	0.06 ± 0.00	0.10 ±0.00

* Results are expressed as mean values ± standard deviation of the mean. DW, Dry weight.

**Table 4 animals-13-00823-t004:** Laminarin and fucoidan content of *L. digitata* and *L. hyperborea* extracts.

*Laminaria* spp. Extract	Laminarin (mg Laminarin/100 mg Freeze-Dried Extract) *	Fucoidan (mg Fucoidan/100 mg Freeze-Dried Extract) *
LDE1	0.52 ± 0.06	4.46 ± 0.03
LDE2	0.70 ± 0.07	3.84 ± 0.06
LDE3	0.44 ± 0.03	5.80 ± 0.05
LDE4	0.67 ± 0.08	5.74 ± 0.05
LHE1	4.94 ± 0.20	14.41 ± 0.46
LHE2	7.59 ± 0.02	12.76 ± 0.34
LHE3	6.17 ± 0.03	14.53 ± 0.12
LHE4	6.19 ± 0.03	14.68 ± 0.37

* Results are expressed as mean values ± standard deviation of the mean. LD1–4, *L. digitata* extract 1–4; LH1–4, *L. hyperborea* extract 1–4.

**Table 5 animals-13-00823-t005:** Effects of seaweed species and harvest season on the selected bacterial populations of the faecal microbiota in the batch fermentation assay (least-square means and standard error of the means).

Bacterial Group (logGCN/g Faeces)	Whole Seaweed Biomass samples				SEM	*p*-Value		
	LDWB-F *	LDWB-N *	LHWB-F *	LHWB-N *		Species	Season	Species × Season
Total bacteria	9.75	9.69	9.76	9.89	0.057	0.032	NS	0.058
Lactobacilli	8.40	8.50	8.78	8.81	0.041	<0.001	NS	NS
*Bifidobacterium* spp.	6.31 ^c^	6.35 ^c^	4.69 ^b^	3.31 ^a^	0.044	<0.001	<0.001	<0.001
*Enterobacteriaceae*	8.13 ^b^	7.79 ^a^	7.68 ^a^	7.98 ^b^	0.068	0.034	NS	<0.001

* Average of least-square mean values at 0, 1, 2.5 and 5 mg/mL. ^a,b,c^ Mean values within a row with different superscript letter were significantly different (*p* < 0.05). NS, not significant (*p* > 0.10); logGCN/g faeces, log-transformed gene copy number per gram of faeces; SEM, standard error of the means.

**Table 6 animals-13-00823-t006:** Effect of harvest season and seaweed concentration on the selected bacterial populations of the faecal microbiota in the batch fermentation assay (least-square means and standard error of the means).

Bacterial Group (logGCN/g Faeces)	Season								SEM	*p*-Value		
	February				November							
	Concentration (mg/mL)									Season	Concentration	Season × Concentration
	0 *	1 *	2.5 *	5 *	0 ^‡^	1 ^‡^	2.5 ^‡^	5 ^‡^				
Total bacteria	9.55	9.86	9.90	9.72	9.73	9.87	9.84	9.71	0.073	NS	0.003	NS
Lactobacilli	8.50	8.67	8.68	8.53	8.62	8.66	8.69	8.64	0.052	NS	0.041	NS
*Bifidobacterium* spp.	6.53 ^d^	6.59 ^d^	5.95 ^c^	2.93 ^a^	6.64 ^d^	6.55 ^d^	3.15 ^b^	2.98 ^a^	0.056	<0.001	<0.001	<0.001
*Enterobacteriaceae*	7.97	8.03	7.90	7.72	8.01	7.90	7.89	7.75	0.085	NS	0.012	NS

* Average of least-square mean values of LDWB-F and LHWB-F. ^‡^ Average of least-square mean values of LDWB-N and LHWB-N. ^a,b,c,d^ Mean values within a row with different superscript letter were significantly different (*p* < 0.05). NS, not significant (*p* > 0.10); logGCN/g faeces, log-transformed gene copy number per gram of faeces; SEM, standard error of the means.

**Table 7 animals-13-00823-t007:** Effects of seaweed species and extraction conditions on the antibacterial and prebiotic potential of the *L. digitata* and *L. hyperborea* extracts in the pure-culture growth assays (least-square means and standard error of the means).

*Laminaria* spp. Extract	Bacterial Strain (logCFU Difference/mL) *				
	*L. plantarum* ^‡^	*L. reuteri* ^‡^	*B. thermophilum* ^‡^	ETEC ^‡^	*S. typhimurium* ^‡^
LDE1	0.16 ^c^	0.28	−0.21 ^a^	−0.45 ^b^	−0.18 ^b^
LDE2	0.15 ^c^	0.25	0.09 ^b^	−0.08 ^c^	−0.01 ^c^
LDE3	0.02 ^ab^	0.03	0.14 ^bc^	0.02 ^cd^	−0.07 ^bc^
LDE4	0.05 ^bc^	0.08	0.89 ^e^	0.04 ^cd^	−0.03 ^c^
LHE1	−0.07 ^a^	0.32	−0.28 ^a^	−1.02 ^a^	−1.14 ^a^
LHE2	0.06 ^bc^	0.28	−0.24 ^a^	0.01 ^cd^	−0.17 ^b^
LHE3	0.07 ^bc^	0.10	0.30 ^cd^	0.26 ^e^	0.07 ^c^
LHE4	0.28 ^d^	0.26	0.43 ^d^	0.09 ^d^	0.00 ^c^
SEM	0.040	0.058	0.073	0.055	0.052
*p*-value					
Species	NS	0.048	0.001	NS	<0.001
Extraction condition	0.013	<0.001	<0.001	<0.001	<0.001
Species × Extraction condition	<0.001	NS	<0.001	<0.001	<0.001

* Bacterial counts are expressed as the difference between the counts of each bacterial strain for each LD/LH extract concentration and their respective control (0 mg/mL). Bacterial counts at 0 mg/mL were as follows: 7.92 ± 0.030 logCFU/mL for *L. plantarum*, 7.40 ± 0.040 logCFU/mL for *L. reuteri*, 6.49 ± 0.055 logCFU/mL for *B. thermophilum*, 8.46 ± 0.046 logCFU/mL for ETEC and 8.93 ± 0.086 logCFU/mL for *S. typhimurium*. ^‡^ Average of least-square mean values at 0.125, 0.25, 0.5, 1 and 2 mg/mL for each extract. ^a,b,c,d,e^ Mean values within a column with different superscript letter were significantly different (*p* < 0.05). LD1–4, *L. digitata* extract 1–4; LH1–4, *L. hyperborea* extract 1–4.; NS, not significant (*p* > 0.10); CFU, colony-forming unit; SEM, standard error of the means.

**Table 8 animals-13-00823-t008:** Effects of extraction conditions and increasing concentrations of *L. digitata* and *L. hyperborea* extracts on the counts of selected beneficial and pathogenic bacterial strains in the pure-culture growth assays (least-square means and standard error of the means).

Extraction Condition	LD/LH Extract Concentration (mg/mL)	Bacterial Strains (logCFU Difference/mL) *				
		*L. plantarum* ^‡^	*L. reuteri* ^‡^	*B. thermophilum* ^‡^	ETEC ^‡^	*S. typhimurium* ^‡^
E1	2	0.09	0.35	−0.96 ^a^	−4.63 ^a^	−3.12 ^a^
	1	0.16	0.27	−0.44 ^b^	0.17 ^cdef^	−0.14 ^d^
	0.5	−0.03	0.34	−0.04 ^cde^	0.27 ^def^	−0.07 ^de^
	0.25	0.04	0.36	0.15 ^efg^	0.36 ^f^	0.00 ^def^
	0.125	−0.02	0.18	0.05 ^defg^	0.17 ^cdef^	0.03 ^def^
E2	2	0.19	0.47	−0.33 ^bc^	−0.25 ^b^	−0.78 ^b^
	1	0.12	0.34	0.12 ^defg^	−0.04 ^bc^	0.05 ^def^
	0.5	0.07	0.34	0.01 ^def^	0.05 ^cd^	0.09 ^ef^
	0.25	0.08	0.06	−0.19 ^bcd^	0.00 ^c^	0.07 ^def^
	0.125	0.05	0.14	0.03 ^defg^	0.06 ^cde^	0.11 ^ef^
E3	2	0.19	0.24	0.33 ^fgh^	0.07 ^cde^	−0.48^c^
	1	0.10	0.04	0.28 ^efgh^	0.29 ^ef^	0.10 ^ef^
	0.5	0.04	0.03	0.25 ^efgh^	0.19 ^cdef^	0.07 ^def^
	0.25	−0.05	−0.03	0.15 ^efg^	0.12 ^cdef^	0.17 ^f^
	0.125	−0.05	0.05	0.08 ^defg^	0.03 ^cd^	0.14 ^ef^
E4	2	0.15	0.32	0.87 ^j^	0.13 ^cdef^	−0.05 ^def^
	1	0.20	0.24	0.93 ^j^	0.10 ^cde^	0.00 ^def^
	0.5	0.20	0.18	0.67 ^ij^	0.04 ^cd^	−0.01 ^def^
	0.25	0.17	0.07	0.47 ^hi^	0.03 ^c^	0.03 ^def^
	0.125	0.09	0.04	0.35 ^gh^	0.04 ^cd^	−0.04 ^def^
SEM		0.064	0.091	0.115	0.087	0.082
*p*-value						
Concentration		0.012	0.002	0.022	<0.001	<0.001
Extraction condition		0.013	<0.001	<0.001	<0.001	<0.001
Concentration × Extraction condition		NS	NS	<0.001	<0.001	<0.001

* Bacterial counts are expressed as the difference between the counts of each bacterial strain for each LD/LH extract concentration and their respective control (0 mg/mL). Bacterial counts at 0 mg/mL were as follows: 7.92 ± 0.030 logCFU/mL for *L. plantarum*, 7.40 ± 0.040 logCFU/mL for *L. reuteri*, 6.49 ± 0.055 logCFU/mL for *B. thermophilum*, 8.46 ± 0.046 logCFU/mL for ETEC and 8.93 ± 0.086 logCFU/mL for *S. typhimurium*. ^‡^ Average of least-square mean values of LD and LH extracts for each extraction condition. ^a,b,c,d,e,f,g,h,i,j^ Mean values within a column with different superscript letter were significantly different (*p* < 0.05). LD, *L. digitata*; LH, *L. hyperborea*; NS, not significant (*p* > 0.10); CFU, colony-forming unit; SEM, standard error of the means.

## Data Availability

All data supporting the reported results are available in this article.

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
