# Peer review of "In Vitro Evaluation of Brown Seaweed Laminaria spp. as a Source of Antibacterial and Prebiotic Extracts That Could Modulate the Gastrointestinal Microbiota of Weaned Pigs"

_animals, 2023, doi:10.3390/ani13050823_

Round 1

Reviewer 1 Report

The above manuscript entitled, "In vitro evaluation of brown seaweed Laminaria spp. as a source of antibacterial and prebiotic extracts that could modulate the gastrointestinal microbiota of weaned pigs" is well written and has enough novelty to be published. In reviewing this manuscript there are little to no correction to be made. The objectives of the work were stated in the introduction and appropriate methods were used to obtain them. The results were clear and crisp and the discussion was well organized. The authors should be commended on the quality of the submitted document. 

  • Additional comment:

The research answere the question on the use of specific sea weed species and their extract as a prebiotic and anitbiotic for weaning pigs using in vitro methods. The document investigated the effect of species, season as well as concentration on microbes in the faeces as well as in vitro effects on specific pathogenic and commensial bacterial species in the gastrointestinal tract.  

The topic is very original and does identify gaps in research that investigates the use of alternative material to maintain gut health in weaned pigs. These reuslts can be then used in in vivo studies to investigate the effect on live animal (growth, and gut health)

Most of the studies undertaken earlier used some sea weed species as suppliments investigating performance and gut health. However, there was never a comparison of specific species, time of collection (month) and the effects of various extract and concentration of these extracts done in vitro. It adds an important gap in the effect of season and concentration of extacts that can be used as a pre and antibiotic for weaned pigs.

The methodology of the author was very well formatted. The only recommendation is for future work where this experiment can be conducted on live weaned pigs with performance parameter, gut microbiota and gut histology being investigated.

The conclusions are consisted with the data presented and discussed.

The references were very appropriate for this manuscript and the number of references were appropraite for this article.

The tables and figures were well constructed and appropriate.

Author Response

No corrections were suggested by this reviewer.

Reviewer 2 Report

The paper deals with the in vitro evaluation of brown seaweed Laminaria spp. as a source of antibacterial and prebiotic extracts that could modulate the gastrointestinal microbiota of weaned pigs. Introduction and methods are  well written. The results of the obtained research were properly described and discussed and brings some new knowledge.

Author Response

(The authors gave the same response as above.)

Reviewer 3 Report

1. The study is devoted to the topical issue of maintaining animal health and developing safe methods of disease prevention of monogastric animals

2. The topic is original and relevant from the point of view of the use of natural resources in order to increase the body's resistance to a period of stress. The authors presented an original methodological approach

3 The authors of the in vitro results found that the type of algae was a more important factor than the harvesting season. At the same time, extraction conditions also have a greater influence on biological activity than the type of algae. This opens up the prospect of harnessing the resources of the sea for the future of animal health.

4 The methods are quite detailed and do not require further development.

5. Conclusions and conclusions about the research problem. The goals set in the publication were achieved and reasonably well argued.

6. All references are appropriate, authors require limited self-citations, which were mostly used in planned methodological approaches (16, 17, 18) or previous studies  (33, 34).

7. To improve the quality of the publication, it is necessary to clarify:

-Clarify the notes to table 1.

-The notes to Table 3 indicate «The units are expressed as  % dry weight (DW) basis». This information is duplicated in the data for the table.

-The notes to Table 4  indicate «Results are expressed as mean values ± standard deviation of the mean». Only mean values are presented.

Author Response

Point 1:

Clarify the notes to table 1.

Response 1:

The footnote was edited as follows ‘Extraction methods and conditions used to produce L. digitata and L. hyperborea extracts as described by Garcia-Vaquero, et al. [17]. LDE1-4, L. digitata extract 1-4; LHE1-4, L. hyperborea extract 1-4; HAE, hydrothermal-assisted extraction; N/A, not applicable’ (Line 186-188).

Point 2:

The notes to Table 3 indicate «The units are expressed as  % dry weight (DW) basis». This information is duplicated in the data for the table.

Response 2:

This information was removed from the footnote and is only included in Table 3 (Line 527).

Point 3:

The notes to Table 4 indicate «Results are expressed as mean values ± standard deviation of the mean». Only mean values are presented.

Response 3:

Thank you for this comment. We have included the above-mentioned missing values in Table 4 of the manuscript (Line 542-545).

Reviewer 4 Report

The topic is interesting for the swine industry and has scientific relevance. This research can contribute to and validate insights about brown seaweed effect on gastrointestinal microbiota. In general, the methods are appropriate for the objectives. The discussion touches on the main findings and their interpretation. In general, the topics chosen for discussion are adequate and interesting.

Furthermore, it would be interesting for the authors to dedicate in the text to the discussion (in 1 paragraph) the choice of these specific algae over others and if In vivo they have already been used to evaluate other effects.

The manuscript requires work before being ready for publication, including specific line comments:

Line 24: Is it appropriate to consider the seaweed extract as therapeutic agent? could it be considered preventative?

Lines 27-28: acronyms should be defined (e.g WB and E1-4 )

Line 109: How these two months were selected?

Line 110: How was seaweed cultivated? Can cultivation methods influence the study, in terms of biological characteristics?

Line 116: having evaluated the direct effect on the faeces, could there be changes in the quality of the algae actions if ingested?

Line 186: missing a 1 μl

Line 192: What is the concentration of agarose gel?

Line 213: specify the composition of the medium used

Lines 237-256: indicating only PROC GLM is not completely clear; you should put the tests in brackets. Which method was used to analyze multiple comparisons should be stated. Multiple comparisons among different treatments were analyzed by LSD, Duncan or other methods?

Lines 387, 395,399, 403: the sentence is a bit dispersive; reverse the sentence and go directly to the point without putting "there was"

Lines 430-442: the results are repeated without giving any justification to the data.

Author Response

Point 1:

Line 24: Is it appropriate to consider the seaweed extract as therapeutic agent? could it be considered preventative?

Response 1:

Thank you for your suggestion. We agree that the term ‘preventative’ is more appropriate, as seaweeds and their extracts are being evaluated in this work and in the relevant scientific literature for the prevention of the negative impact of weaning in the gastrointestinal health of pigs. The manuscript was edited accordingly (Line 24).

Point 2:

Lines 27-28: acronyms should be defined (e.g WB and E1-4)

Response 2:

The manuscript was edited to better define the acronyms used as follows ‘Particularly, February and November whole seaweed biomass samples of L. hyperborea (LHWB-F and LHWB-N) and L. digitata (LDWB-F and LDWB-N) were used. In the next part of the study, the increasing concentrations of four extracts produced from L. hyperborea (LHE1-4) and L. digitata (LDE1-4)……’ (Line 27-30).

Point 3:

Line 109: How these two months were selected?

Response 3:

Thank you for this comment. These two months were selected based on the seasonality that characterise the concentration of seaweed polysaccharides. Particularly, autumn and winter are associated with high and low polysaccharide content, respectively. This is attributed to the accumulation of the polysaccharides in the autumn which are then being utilised for survival during winter. Thus, November and February were selected as representative months for each of these seasons.

Point 4:

Line 110: How was seaweed cultivated? Can cultivation methods influence the study, in terms of biological characteristics?

Response 4:

Thank you for these interesting questions. Wild seaweeds were evaluated in the current study and, thus, no cultivation methodology was used. We would expect that the cultivation methods would influence the biological characteristics of the seaweed components as these are influenced by the environmental conditions. However, answering those questions is beyond the scope of the current study.

Point 5:

Line 116: having evaluated the direct effect on the faeces, could there be changes in the quality of the algae actions if ingested?

Response 5:

Thank you for this comment. Seaweed polysaccharides are non-digestible compounds reaching the colon with their chemical structure intact. Therefore, their ingestion would not be expected to alter the biological activities.

Point 6:

Line 186: missing a 1 μl

Response 6:

In Line 186, it was stated: ‘1 μL of each primer’ meaning 1 μL of forward primer and 1 μL of reverse primer which along with the 3 μL template DNA, 5 μL nuclease-free water and 10 μL of Fast SYBR® Green Master Mix added up to the final 20 μL reaction volume. To avoid any confusion, we have changed ‘1 μL of each primer’ to ‘1 μL of forward primer (10 μM), 1 μL of reverse primer (10 μM)’ in the manuscript (Line 236-237).

Point 7:

Line 192: What is the concentration of agarose gel?

Response 7:

A 2% agarose gel was used for the evaluation of the PCR products. This information has been included in the manuscript (Line 416).

Point 8:

Line 213: specify the composition of the medium used

Response 8:

The 10% medium used in the pure culture growth assays was either 10% de Man, Rogosa and Sharpe broth for L. plantarum, L. reuteri and B. thermophilum cultures, or 10% Tryptone Soya Broth for S. Typhimurium and ETEC cultures. Both media were purchased from Oxoid and are of known composition. The relevant sentence was edited in the manuscript to avoid any confusion for the reader (Line 436-438).

Point 9:

Lines 237-256: indicating only PROC GLM is not completely clear; you should put the tests in brackets. Which method was used to analyze multiple comparisons should be stated. Multiple comparisons among different treatments were analyzed by LSD, Duncan or other methods?

Response 9:

In the statistical analysis of the data collected from the batch fermentation and pure culture growth assays, differences between least square means were investigated using the t-test after Tukey’s adjustment for multiple comparisons. The used method was also included in brackets at the relevant parts of the manuscript (Line 498, 507).

Point 10:

Lines 387, 395,399, 403: the sentence is a bit dispersive; reverse the sentence and go directly to the point without putting "there was"

Response 10:

We thank the reviewer for his suggestion. We agree that such sentences can be dispersive. However, in the case of our manuscript, our statistician believes that it is important to state the interaction or effect identified during the statistical analysis of the data when presenting the results for each bacterial group or species, as it could be helpful for the readers. For this reason, we would rather keep the suggested sentences or the ones prior to them in the results section of the manuscript, if possible.

Point 11:

Lines 430-442: the results are repeated without giving any justification to the data.

Response 11:

Thank you for this comment. In the above-mentioned lines, we have provided the readers with a brief summary of the main findings from each experimental step of this study. This overview sets the scene for the more detailed discussion of the different observations reported. We would like to keep this format.

Point 12:

It would be interesting for the authors to dedicate in the text to the discussion (in 1 paragraph) the choice of these specific algae over others and if in vivo they have already been used to evaluate other effects.

Response 12:

We thank the reviewer for his/her kind suggestion. Regarding the selection of Laminaria spp. over other seaweed species, we have already explained in the introduction that this seaweed has previously been investigated as a dietary supplement to beneficially modulate the gastrointestinal microbiota in pigs (Line 91-100). We therefore considered it a suitable candidate for producing the extracts evaluated in the current study. As far as the other biological activities of Laminaria spp. are concerned, we have included a few lines in the introduction with relevant information on these (Line 88-91).